# External Inflammatory Root Resorption in Traumatized Immature Incisors: MTA Plug or Revitalization? A Case Series

**DOI:** 10.3390/children10071236

**Published:** 2023-07-18

**Authors:** Tchilalo Boukpessi, Leslie Cottreel, Kerstin M. Galler

**Affiliations:** 1Department of Restorative Dentistry and Endodontics, Faculty of Dentistry, University Paris Cité, 1 rue Maurcice Arnoux, 92120 Montrouge, France; dr.cottreel@gmail.com; 2Pitié Salpétrière Hospital Assistance Publique-Hôpitaux de Paris, 75013 Paris, France; 3Laboratory of Biomedical Research in Odontology, URP 2496, Faculty of Dentistry, University Paris Cité, 1 rue Maurice Arnoux, 92120 Montrouge, France; 4Private Practice, 10 rue bis Madame, 78000 Versailles, France; 5Department of Operative Dentistry and Periodontology, University Hospital Erlangen, University of Erlangen-Nuremberg, Glückstr. 11, 91054 Erlangen, Germany; kerstin.galler@uk-erlangen.de

**Keywords:** immature teeth, dental trauma, pulp necrosis, MTA plug, revitalization, case series

## Abstract

Introduction: External inflammatory root resorption (EIRR) in immature permanent teeth is a common complication after severe dental trauma. The management of this condition requires thorough disinfection of the root canal in order to arrest the resorptive process. However, current guidelines regarding the recommended treatment of EIRR following traumatic dental injuries vary, mainly in regard to the type of intracanal medication and its retention time in the root canal system. The objective of this case series was to present both the apical barrier technique (MTA plug) and revitalization procedures as valid treatment options in immature teeth with EIRR. Methods: Four cases of post-traumatic immature teeth diagnosed with pulp necrosis and EIRR, with or without apical periodontitis, were treated either by an MTA plug (two teeth) or revitalization (two teeth). Cases were followed between 12 and 24 months. Results: Both treatment methods were efficient in arresting EIRR and enabled bone healing. After revitalization, partial root maturation was observed. Conclusion: Whereas the key to achieve periodontal healing in cases of EIRR is thorough disinfection of the root canal, both a subsequent MTA plug as well as revitalization may represent adequate treatment methods. An additional benefit lies in the potential of revitalization to promote further root maturation through hard tissue apposition.

## 1. Introduction

### 1.1. External Inflammatory Root Resorption after Dental Trauma

Pathological resorptive processes in permanent teeth result in the destruction of mineralized root structures, namely cementum and dentin, due to the action of clastic cells. Several different types of resorptions can be distinguished. Rapidly progressing external inflammatory (infection-related) root resorption (EIRR) is a commonly found complication after dental trauma, in particular after luxation or avulsion. A recent systematic review and meta-analysis by Souza et al. reported an incidence of 23.2% for EIRR after avulsion injuries [1].

The impact of the injury is two-fold: the damage to root cementum and periodontal ligament cells covering the root surface results in the loss of a protective barrier, which prohibits root resorption under physiological conditions. Necrosis of the dental pulp as a consequence of tooth displacement allows for subsequent population of the root canal with microorganisms. This microbial contamination induces an immune reaction resulting in root resorption [2,3,4,5]. The destruction of root cementum leads to exposure of the dentine on the outer root surface, which makes this tissue accessible to clastic cells. These can bind exclusively to mineralized tissue surfaces to initiate the resorptive process [6], which starts as a transient surface resorption. In cases where endodontic treatment is not initiated within the first 2 weeks after the traumatic incident, microbial contamination of the root canal occurs. Subsequently, bacterial toxins diffuse through the dentinal tubules towards the root surface and trigger, stimulate, and amplify the ongoing resorption [3]. At later stages, periapical radiographs show periradicular radiolucent areas, suggesting active resorption; however, the canal remains clearly identifiable. Whereas initial radiographic signs of EIRR may be visible already two weeks after replantation of avulsed teeth [7,8], the limitations of periapical radiographs have been well reported, associated with a potentially late diagnosis which compromises prognosis [9]. Cone beam computed tomography (CBCT) has demonstrated reasonable diagnostic ability and can reveal the presence of a localized perforation of the canal. Earlier detection of resorptions may allow faster and more appropriate treatment, thus leading to an improved prognosis of traumatized immature teeth. Recently, the European Society of Endodontology published a position statement, stipulating that prior to potential management of EIRR, CBCT imaging may be a valid tool to assess the nature of the resorptive process and to detect root canal perforations, which could easily be missed by conventional radiographs [10].

### 1.2. Treatment Options

Conventionally, based on Cvek’s study, long-term calcium hydroxide (CH) intracanal dressings have been the treatment of choice in cases of EIRR [11]. The use of CH as intracanal medication can result in an arrest of the resorptive process and promote the healing of periodontal lesions [12]. The recommendations for the duration of CH intracanal medication in EIRR vary from 4 weeks to several months [13,14]. The guidelines of the International Association of Dental Traumatology (IADT) recommend the use of CH for 3 weeks and subsequent replacement every 3 months until radiolucencies caused by the resorptive lesion disappear [15]. This use of long-term CH dressing requires greater patient compliance and more time and efforts from the dental surgeon [11]. An additional drawback is the increased risk of root fracture, as CH degrades the collagenous matrix of dentine, resulting in its reduced biomechanical stability [16]. Currently, guidelines regarding the treatment of EIRR as a consequence of traumatic dental injuries vary, mainly regarding the type of medication and its retention time within the root canal system as well as the type of final obturation [10,15]. If root formation in the respective teeth is incomplete, this poses additional challenges to the practitioner.

To date, three treatment options for immature necrotic teeth are available, which are (1) CH apexification by means of long-term intracanal dressing with CH; (2) an apical mineral trioxide aggregate (MTA) plug, where a hydraulic calcium silicate cement (MTA) is placed inside the canal in contact with the apical tissues; and (3) revitalization, where provocation of bleeding into the root canal may lead to new tissue formation and healing. Due to the limitations of CH apexification as discussed above, this treatment is no longer considered a first choice. The apical barrier technique with MTA appears successful in promoting hard tissue formation at the apex of immature teeth with pulp necrosis, and also in cases with periapical lesions, after a step of antisepsis [17].

The purpose of revitalization is to replace damaged structures, including dentine and root structures as well as cells of the pulp–dentin complex, in addition to the resolution of apical periodontitis [18]. As a result of new tissue formation inside the root canal, apposition of mineralized tissue onto the existing internal root surfaces is possible and may strengthen fracture-prone immature teeth [19]. This may be beneficial in teeth with early stages of root development, which exhibit particularly thin root walls. Details on the indication, procedure, and expected outcomes have been provided by the European Society of Endodontology [20]. A report by Tzanetakis described an arrested EIRR in a traumatized tooth after revitalization, where the blood clot served as a natural guide rail for repair [21]. More recently, a case series of four teeth showed successful clinical outcomes after the use of platelet-rich fibrin (PRF), which led to an arrest of trauma-induced EIRR [22].

Due to the rather limited amount of data in the literature on the proposed subject, the aim of this article was to assess the outcome after MTA plug or revitalization in four infected immature permanent teeth with EIRR after dental trauma in terms of arrest of the resorptive process, periapical bone healing, root development, and pulp vitality.

## 2. Materials and Methods

### 2.1. Patients

The patients were referred to the post-traumatology consultation office of the Department of Oral Medicine at Pitié Salpétrière-Charles Foix (APHP France). During the first session, clinical and radiographic examination was performed along with clinical tests, including cold and electric pulp testing (EPT) [23], percussion, tooth discoloration, and assessment of periodontal conditions (pocket depth and mobility). A periapical radiograph was taken, and CBCT imaging [10] was performed pre-operatively, followed by treatment planning. Subsequently, either an MTA plug or revitalization was chosen as the therapeutic intervention, and the patient and parents were provided with both general and specific information regarding the following:-The existing condition;-MTA plug and revitalization, with their potential advantages, disadvantages, and present gaps in knowledge;-Duration of treatment and follow-ups;-Use of materials and medicaments;-Treatment alternatives;-Potential outcomes.

Informed consent was obtained from the patients’ parents since all patients were minors. As both the MTA plug and revitalization are valid treatment options for immature teeth with pulp necrosis and recommended by Endodontic Societies, no further ethical considerations were necessary.

### 2.2. Clinical Protocol for the MTA Plug

The main steps of the MTA plug protocol during the first appointment included local anesthesia (Septanest 40 mg/mL adrenaline 1/200,000, Septodont, Saint Maur des fossés, France), isolation of the tooth with a dental dam, preparation of an access cavity, root canal exploration, determination of working length, root canal disinfection by irrigation with sodium hypochlorite (3%) and subsequent activation (EndoActivator, Dentsply Sirona, Ballaigues, Switzerland), removal of necrotic tissue, and application of intracanal dressing with CH. The second appointment took place three weeks later, where the absence of signs of inflammation was evaluated. After local anesthesia and isolation with a dental dam, the canal was irrigated with sodium hypochlorite (3%) for disinfection purposes and to remove CH remnants. The root canal was dried with paper points. Then, a 4–5 mm MTA (PROROOT MTA white, Dentsply Sirona, Ballaigues, Switzerland) apical plug was placed, followed by radiographic control. During the third appointment, the rest of the canal was filled with warm gutta-percha, and an adhesive coronal restoration was placed.

### 2.3. Clinical Protocol for Revitalization

The clinical protocol for revitalization according to the European Society of Endodontology guidelines [20] was implemented. Local anesthetic with adrenaline (SEPTANEST 40 mg/mL ADRENALINE 1/200,000, Septodont, Saint Maur des Fossés, France) was injected, and the tooth was isolated with a dental dam. The access cavity was re-opened, and the canal was irrigated with sodium hypochlorite (3%, 20 mL) using a side-vented needle, placed 2 mm above vital tissue (by use of an operating microscope). Then, the canal was dried with sterile paper points and rinsed with 20 mL of EDTA (17%). A commercially available non-discoloring preparation of CH was inserted evenly into the canal. A temporary coronal seal with a glass ionomer cement was placed directly onto the intracanal dressing. Two weeks later, during the second appointment, an absence of signs of inflammation was confirmed. Local anesthetic without a vasoconstrictor was administered, and the tooth was isolated with a dental dam. After the removal of the glass ionomer seal, the canal was rinsed with EDTA (17%, 20 mL, 5 min) using a side-vented needle placed 2 mm above the vital tissue (Figure 1A). Subsequently, the canal was irrigated with sterile saline (5 mL) to diminish undesirable effects of irrigants on the target cells. Excess liquid was removed by use of paper points. Bleeding was induced by introducing a pre-bent Hedström file size 40 into the canal and performing a rotational movement to disrupt the apical tissue. Bleeding was provoked until 2 mm below the cemento-enamel junction (Figure 1B). A hydraulic calcium silicate cement (Biodentine, Septodont, Saint Maur des fossés, France) was placed on top of a collagen scaffold (Collacone Botiss, Strauman, Basel, Switzerland) (Figure 1C,D). The cement was used as a temporary restorative material (Figure 1E). At a later session, composite resin was placed as a final restoration material.

### 2.4. Follow-Up

Periapical radiographic and small field-of-view cone beam computed tomography (CBCT) (PHT-6500; Vatech Co, Ltd., Gyeonggi-do, Republic of Korea (90 kV and 7.0 mA)) images were taken before treatment. Patients were recalled for examinations after 3, 6, 12, and 24 months. During the follow-up visits, clinical parameters including pain, swelling, sinus tract, mobility, crown discoloration, and pulp sensibility were assessed. A positive response to pulp sensibility testing after revitalization was documented only if the teeth responded repeatedly and reliably to cold testing and/or electric pulp testing. A periapical radiograph was taken at each recall session to evaluate bony healing, the arrest of EIRR and periodontal healing for all teeth, and the potential root development in teeth after revitalization.

Cases were considered failures if one of the following criteria was fulfilled: clinical symptoms (pain, swelling, or sinus tract); no change in root length; or signs of apical closure, recurrence of apical periodontitis, and external root resorption.

## 3. Results

### 3.1. Case Report 1

An 8-year-old girl was referred by a general practitioner to an endodontic specialty practice after an EIRR was diagnosed in teeth 11 and 21. The girl’s parent reported a history of dental trauma eight months earlier. Tooth 11 had been avulsed and replanted, where storage conditions remained unclear. Tooth 21 had been luxated. The medical history was unremarkable. Clinical examination showed no pain upon tissue palpation and no reaction to cold testing. Periodontal probing depths were below 3 mm. Both teeth 11 and 21 were tender to percussion. Periapical radiographs revealed large open apices on both teeth and the presence of periapical radiolucencies and multiple radicular resorption lacunae. Advanced resorption lacunae on the distal aspect of the root were observed for tooth 11 (Figure 2A). CBCT images revealed the presence of resorption on the palatal area of both teeth (Figure 2B–E). In summary, pulp necrosis with symptomatic apical periodontitis was diagnosed, and MTA plug apexification was chosen for both teeth. The clinical protocol for the apical barrier technique was implemented as described above (Figure 2F). At the 3-month follow-up appointment, both teeth were asymptomatic and tested negative in all clinical tests. At the 12-month follow-up examination, both teeth continued to be asymptomatic and revealed a negative response to thermal testing and percussion (both reaction and sound), and palpation responses were within normal limits. Furthermore, resorption lacunae were already arrested, and complete bony healing was observed on periapical radiographs and CBCT images (Figure 2G–I).

### 3.2. Case Report 2

An 8-year-old boy was referred by a general practitioner to an endodontic specialty practice after external root resorption was diagnosed in tooth 11. Two months before, tooth 11 was replanted 2 h after avulsion; it had been stored in milk immediately after the injury. The parents also reported crown fractures of teeth 11 and 21. Clinical examination showed a buccal fistula, mobility of tooth 11, and a negative response to pulp sensitivity testing. Periapical radiographs revealed a wide-open apex on tooth 11 and the presence of a periapical radiolucency and multiple radicular resorption lacunae (Figure 3A). The presence of a periapical radiolucency and marked bone loss around the tooth were observed (Figure 3B,C). Altogether, a diagnosis of necrotic pulp with symptomatic apical periodontitis was made. The prognosis was assessed as unfavorable, but the alternative treatment of extraction would result in subsequent bone loss and severely compromised options for rehabilitation. Therefore, the decision to try to keep the tooth was taken even though the prognosis was uncertain. The apical barrier technique (MTA plug) was selected as the treatment of choice at this stage. The clinical protocol was performed as described above. After temporary intracanal dressing with CH (Figure 3C), an MTA plug was placed (Figure 3D). Then, teeth 11 and 21 were restored coronally with a resin composite. At the 3-month follow-up appointment, both teeth were asymptomatic, and the fistula had disappeared. Teeth 11 and 21 tested negatively in all clinical tests. The periapical radiograph showed resolution of the periapical inflammation and an arrested resorption process (Figure 3E). At the 12-month follow-up examination, both teeth continued to be asymptomatic and revealed a negative response to thermal, percussion, and palpation tests. Furthermore, the resorption was arrested, and complete bony healing was observed on a periapical radiograph (Figure 3F).

### 3.3. Case Report 3

A girl aged 7 years and 10 months had experienced a trauma involving avulsion of tooth 21. The tooth was replanted at the Emergency Department at Pitié Salpétrière Hospital. Three months later, the patient’s chief complaint was mobility of the avulsed and replanted maxillary left central incisor. Clinical examination showed mobility of tooth 21 and a negative response to pulp sensitivity testing. A periapical radiograph revealed a wide-open apex on tooth 2 and the presence of a periapical radiolucency and multiple radicular resorption lacunae (Figure 4A). CBCT images showed external root resorption without root perforation (Figure 4B–D). Pulp necrosis with symptomatic apical periodontitis associated with EIRR was diagnosed. In this case, revitalization was chosen and performed as described above (Figure 4E). During the first follow-up appointment, tooth 21 responded negatively on palpation and weakly to percussion. Electric and thermal sensitivity tests were negative. Mobility and periodontal pocket depths were normal for all teeth. At the 6-month follow-up, no pain to percussion was noted, and electric and thermal tests were positive but delayed when compared to adjacent teeth. No discoloration was observed. Radiographically, a decrease in the apical radiolucency, an increase in root length, and a decrease in the size of the apical foramen were observed (Figure 4F). At the one-year follow-up, no pain was reported, electric and thermal testing evoked a positive response, and mobility and periodontal tissues were normal. Interestingly, on the periapical radiograph and on CBCT images, the periapical lesion still decreased, the resorption was stopped, and apical closure was evident when compared with previous radiographs (Figure 4G–J).

### 3.4. Case Report 4

A 7-year-old boy had experienced a trauma and avulsed tooth 21. A few weeks after replantation, the dental pulp had become necrotic, and the periapical radiograph revealed multiple resorption lacunae on the distal side. In addition, a wide-open apex and a periapical radiolucency were detected. CBCT was performed and revealed that even though there were multiple lacunae, no perforation of the canal walls was visible (Figure 5A–D). Based on these findings, a diagnosis of pulp necrosis with symptomatic apical periodontitis was made. Then, revitalization was chosen for the treatment (Figure 5E). Three months after treatment, a failure was noted due to the presence of a fistula without any changes on the periapical radiograph (Figure 5F). Then, a decision was made to perform an MTA plug procedure (Figure 5G,H). During the follow-up sessions (6 months and 1 year, Figure 5I,J, respectively), the tooth had survived and was asymptomatic, and the apical lesion had disappeared. Tooth 21 showed an increased root length and apical closure by the formation of a calcified barrier (Figure 5J).

## 4. Discussion

Among dental traumatic injuries, the most severe are intrusions and avulsions. Frequently, the replantation of avulsed or intruded teeth leads to root resorption due to the severity of the traumatic impact [24]. Two types of resorptions may develop after these types of injuries, namely external inflammatory root resorption (EIRR) and replacement resorption (RR) [3]. A combination of loss of barrier function at the root surface and post-traumatic infection of the root canal leads to EIRR. This type of resorption can be prohibited by early initiation of root canal treatment. If clinicians fail to do so and the resorptive process is diagnosed later on, it is still possible to arrest it by thorough disinfection. RR, on the other hand, occurs after severe damage to cementum affecting more than 20% of the root surface, which leads to extensive cell necrosis. Repair mechanisms fail and the injury cannot heal [25]. As a result, radicular dentine falls prey to the regular bone remodeling process, where it is slowly resorbed and replaced with bone. RR can evolve without additional (infection-related) factors, and no treatment approach is available to date to prevent or arrest it. The primary damage to the root surface is not extensive after avulsion; however, replacement resorption is likely to occur if the tooth was not stored under favorable conditions, particularly dry storage for more than 60 min [25,26]. In that case, periodontal ligament cells covering the root become necrotic, and healing of the periodontal ligament after replantation is no longer possible. A recent systematic review by Souza et al. showed that in case of avulsion and replantation, the incidence of replacement resorption amounts to 51.0% [1]. Both EIRR and RR may occur in the same tooth if the abovementioned conditions are met.

### 4.1. Management of EIRR

In cases of EIRR, the dental surgeon must first manage the infection by setting up optimal disinfection of the root canal to eliminate pro-inflammatory microbial stimuli. However, the clinician must also consider the risk of development of RR based on the type and severity of the injury, which plays a role for the long-term prognosis of the respective tooth. The question may arise of whether an MTA plug or revitalization will be the best treatment option, as they are both adequate procedures for immature permanent teeth with pulp necrosis and apical periodontitis. The success rates of revitalization and MTA plug procedures have been reported to be similarly high [27]. Currently, there are no specific recommendations or clear guidelines regarding the most favorable treatment for EIRR on lateral root surfaces following dental trauma. The main challenges faced by the practitioner consist of (1) arresting inflammatory resorption by disinfection and elimination of bacteria and bacterial toxins from the root canal and (2) healing apical periodontitis. The promotion of root formation through the increase in wall thickness and in root length, which may be achieved only by revitalization, could be beneficial, but may not be a priority at this first stage.

When EIRR is observed, traditionally, in addition to sodium hypochlorite irrigation, which efficiently reduces the bacterial load, long-term CH has been the treatment of choice [10,11,13,15]. This intracanal medication may arrest resorption and induce the formation of a mineralized barrier at the apex, against which a root canal filling with warm gutta-percha is condensed. Due to the abovementioned drawbacks, it seems appropriate to prefer the use of MTA to induce an apical barrier. Furthermore, a recent systematic review and meta-analysis stated lower success rates for CH apexification compared to the MTA plug [28]. In the cases described here, CH was used for canal disinfection for at least 3 weeks to optimize intracanal disinfection. Then, an apical plug of MTA was placed under an operating microscope. The ability of MTA to form a sufficient plug is a critical factor for apical healing. The first two clinical cases presented here, treated with an MTA plug, showed a favorable outcome after one year. Symptoms of inflammation disappeared rapidly after the treatment, and bone healing was visible on periapical radiographs and CBCT images (case 1). It is important to highlight that the guidelines (the American Association of Endodontists and the American Academy of Oral and Maxillofacial Radiology joint statement) should be followed, and routine use of CBCT imaging on patients should be avoided. In that first case, the patient experienced a new trauma one year after treatment, which provided the reason to take CBCT imaging to exclude bone fractures.

### 4.2. Subsequent Treatment by MTA Plug or Revitalization

The MTA plug procedure is the conventional method for the treatment of young, permanent, non-vital teeth. In a systematic review, Duggal et al. suggested the use of MTA followed by root canal obturation as the treatment of choice for the endodontic management of traumatized immature permanent anterior teeth in children and adolescents, where a risk of replacement resorption could compromise prognosis [29]. In a recent prospective randomized controlled trial comparing revitalization and MTA plug in immature necrotic teeth, the results showed that the overall success rate was 97% in the MTA group and 89.8% in the revitalization group [28]. However, an advantage of revitalization over the MTA plug was noticed because the former may result in increased root length and thickness. It has been hypothesized that stem cells from the apical papilla (SCAPs), which are present in immature teeth, are transported into the root canal with the induction of bleeding during revitalization, and may contribute to new tissue formation and healing [30]. It has recently been shown that these stem cells are able to survive and retain their stemness despite an ongoing inflammatory status [31,32]. Moreover, SCAPs and other stem cell populations have been shown to have immunomodulatory effects [32].

Both interventions aim at the survival of immature necrotic teeth. One of the first studies observing the arrest of EIRR treated by means of revitalization used a triple antibiotic paste as intracanal medication (not CH), and the treatment was successful in arresting resorption and in healing apical periodontitis [33]. In 2016, Saoud et al. described a series of three cases of traumatized teeth with EIRR treated by revitalization, where the resorption was arrested after treatment in all cases [34]. More recently, Tzanetakis described a case in which revitalization arrested EIRR in a tooth after an intrusive injury. During the 30-month follow-up period, progressive healing and remodeling of the bone were detected [21]. It has been shown that the use of CH as intracanal medication from the first appointment can arrest periradicular resorption and promote healing [21]. In the third case shown here and treated by revitalization, the resorption was stopped and the root continued its development, a process that depends on sound surrounding tissue structures. During a revitalization procedure, the use of EDTA and calcium hydroxide enables the exposure of collagenous fibers due to dentine demineralization, which can promote cell adhesion [35]. Furthermore, the release of embedded bioactive molecules from the dentine matrix facilitates the proliferation and differentiation of cells [35]. Both factors might contribute to new tissue formation. A hypothesis by Yoshpe et al. stated that the release of bioactive and chemotactic molecules will, among other effects, lead to changes in local pH to promote stem cell migration into the root canal and onto affected internal root surfaces to initiate the repair process [22]. In the same study, the authors showed, by means of four clinical cases, the arrest of root resorption after revitalization with PRF (platelet-rich fibrin). For all cases of the present report, no PRP (platelet-rich plasma) or PRF was used—only a blood clot in combination with a collagen matrix as a cover. Other studies have shown that PRF or PRP could be better methods than an induced blood clot as they contain additional cells and growth factors [36,37].

### 4.3. Discussion of Methodology

In evidence-based medicine (EBM), case series have the least robust study designs. Thus, they are not fully considered and used for decision-making in medicine and dentistry. In a recent study aiming to describe the contextual elements that led to EBM, the authors concluded that EBM needs to develop further and realize that health and healthcare rely on complex interactions. Therefore, it may be advantageous to analyze a multitude of scientific data [38]. Therefore, if a careful approach has been used to collect data on patients, i.e., case selection and outcome analysis, the results can supply valuable insights and might be used to deduce treatment decisions. In the present study, careful patient selection, meticulously performed treatment procedures according to the guidelines of Endodontic Societies, and thorough follow-ups were conducted, therefore ensuring high-quality data for further analysis, which may contribute to our understanding of outcomes in these complex cases.

### 4.4. Conclusions

In conclusion, both revitalization and the MTA plug are equally suitable interventions which result in similar overall outcomes. Although the MTA plug shows slightly higher success rates, revitalization can promote an increase in root length and closure of the apical foramen and thus strengthen immature teeth. The cases presented here emphasize this. An MTA plug can still be performed if revitalization as a less invasive treatment option fails, with persisting symptoms or apical lesion. Therefore, this case series confirms that revitalization is an alternative treatment option for immature teeth with EIRR; however, the need for well-designed clinical studies is obvious.

## Figures and Tables

**Figure 1 children-10-01236-f001:**
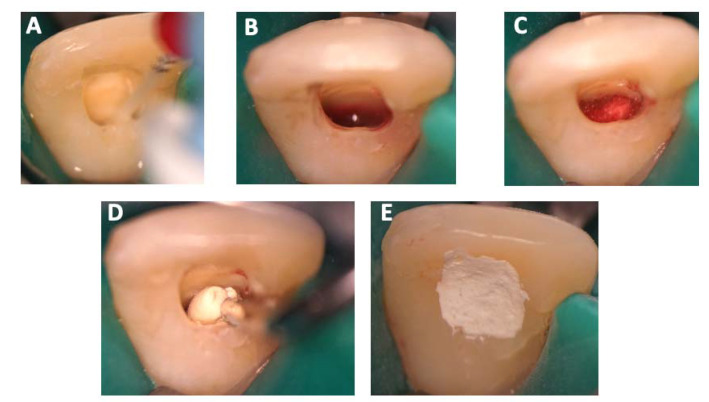
Second session of revitalization, clinical images. (**A**) Removal of CH intracanal dressing. (**B**) Bleeding was provoked up to 2 mm below the gingival margin. (**C**) Placement of a collagen matrix to cover the blood clot. (**D**) Application of hydraulic calcium silicate onto the collagenous matrix. (**E**) Placement of the temporary material.

**Figure 2 children-10-01236-f002:**
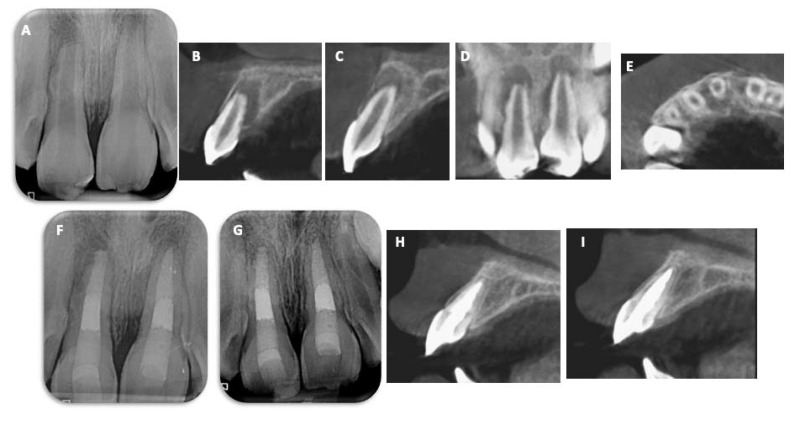
Case 1: Apexification with MTA plug. (**A**) Preoperative intraoral radiograph of teeth 11 and 21 presenting apical lesion and EIRR. (**B**–**E**) Sagittal, frontal, and axial CBCT images show the presence of resorption on all aspects of teeth 11 and 21. (**F**) Postoperative radiograph after MTA plug and filling with gutta-percha. The root canal filling was performed 1 week after the placement of MTA. At the 12-month follow-up, resorption lacunae were arrested, and complete bone healing was observed on periapical radiographs (**G**) and CBCT images (**H**,**I**).

**Figure 3 children-10-01236-f003:**
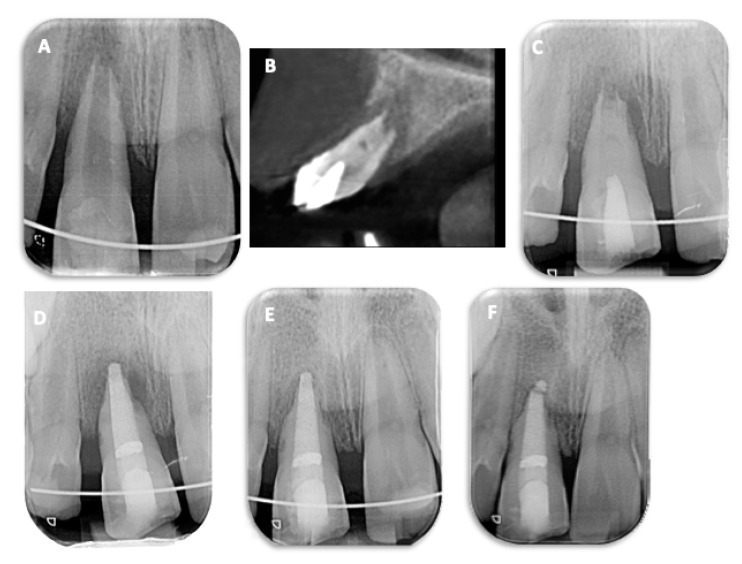
Case 2. (**A**) Periapical radiograph reveals a wide-open apex on tooth 11 and the presence of a periapical radiolucency and multiple radicular resorption lacunae. (**B**) The presence of a periapical radiolucency and relevant bone loss around the tooth are observed on CBCT sagittal images. The clinical protocol for an MTA plug was performed. (**C**) Temporary intracanal dressing with CH before MTA plug in (**D**). (**E**) At the 3-month follow-up, the periapical radiograph showed resolution of the periapical inflammation and an arrested resorption process. (**F**) At the 12-month follow-up, the resorption was arrested, and complete bone healing can be observed on the periapical radiograph.

**Figure 4 children-10-01236-f004:**
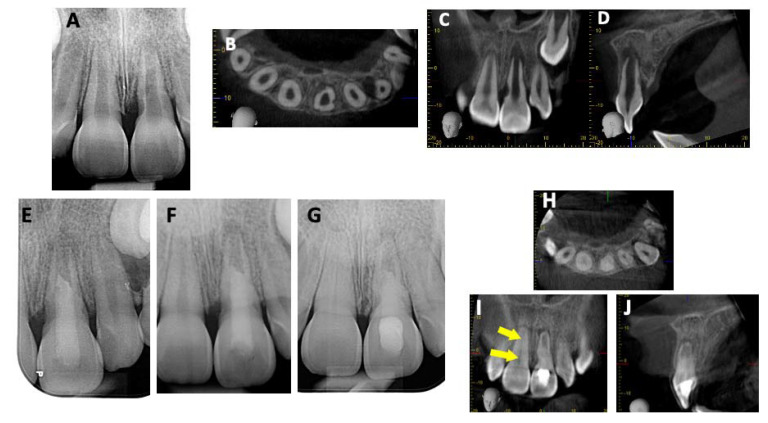
Case 3: Revitalization treatment of tooth 21 after avulsion and replantation. (**A**) The periapical radiograph reveals a wide-open apex and the presence of a periapical radiolucency and multiple radicular resorption lacunae. (**B**–**D**) CBCT images show external root resorption on different aspects of the root, without root perforation. (**E**) Post-operative periapical radiograph after revitalization. (**F**) Six-month follow-up radiograph shows a decrease in the apical radiolucency, an increase in root length, and a decrease in apical foramen size. (**G**–**J**) On the periapical radiograph and on CBCT images, the periapical lesion still decreased, the resorption was stopped, and the apex was closed when compared with previous radiographs.

**Figure 5 children-10-01236-f005:**
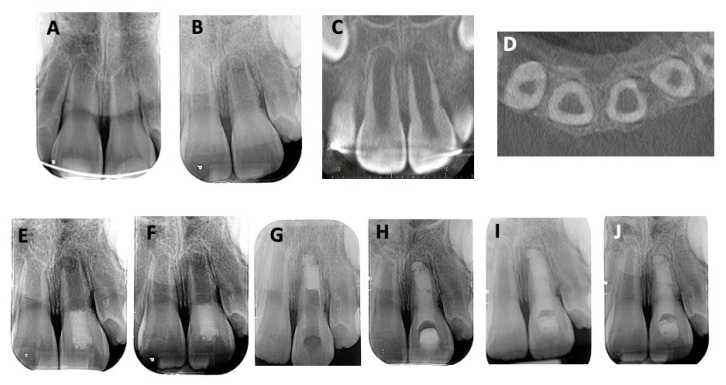
Case 4: MTA plug after failure of revitalization. (**A**,**B**) A periapical radiograph revealing multiple resorption lacunae on the distal side of tooth 21, a wide-open apex, and a periapical radiolucency. (**C**,**D**) CBCT revealed that even though there were multiple lacunae, no perforation of the root walls was visible. (**E**) Revitalization procedure. (**F**) Failure noted due to no decrease in the periapical radiolucency. (**G**) MTA plug. (**H**) Obturation with gutta-percha. (**I**,**J**) Follow-up sessions for respective months and 1 year. Increase in root length and apical closure (**J**).

## Data Availability

All available data regarding these clinical cases coul be asked by email to corresponding author.

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
