# Peer review of "External Inflammatory Root Resorption in Traumatized Immature Incisors: MTA Plug or Revitalization? A Case Series"

_children, 2023, doi:10.3390/children10071236_

Round 1

Reviewer 1 Report

In general, the document can be considered for publication after solving some necessary points in the part of the methodology and bibliographic citations. 

Specifically, it is suggested to adopt and apply the guidelines for clinical case reports, or case reports. This will allow to have a document with more scientific support and a better justification and application of the method. 

The use of The CARE guidelines (for CAse REports) is encouraged, as well as following the checklist sheet, in order to be able to justify one by one the requirement for the application and follow-up of these guidelines. 

In the case of the bibliographic references, some of them are very old, it is understood that they are historical references but it would be worthwhile to keep them and also add some that are more recent and that allow updating the information with the objective of having an updated manuscript and according to what is intended to be presented, highlighting the innovative or interesting part by comparing what we have with what is offered. 

Reviewer 2 Report

Dear authors : the case series is good, but it needs extensive grammatical check. below are some of the examples. Please add studies in the introduction which supports more about the technique. When you talk about the introduction of treatment options, please quote studies which have used these agents and found positive effect

abstract :  line 26 remove the word "for" 

Introduction: dont use complex sentences. split the sentence into tow especially in line 52, the limitations 

is there any specific guidelines by IAPD on EIRR

Methodology : Ethical considerations not mentioned   

                        please give bullets from 109 to115

                        line 129 to 130 - rewrite the sentence 

                        line 169 --- write it as one not 1

Discussion : can be improved, so depending on the case series, discusion can be elaborated quoting the advantages of each agent in the procedure and drawbacks for the same .

English editing by expert required 

Reviewer 3 Report

MTA ARTICLE REPORT. 

 The article presents a very interesting subject with great clinical repercussions for the dentist. In general, it is very well written and meets the requirements of a scientific article. However, it is necessary to point out some considerations. 

The introduction, although very well founded, lacks the treatment proposed by the IADT for this type of problem and should be included in an article on this subject. It cites only in line 69 an article by Bourguignon that is not adequately included in the bibliography. 

He even cites case series as the existing studies on the theme, but there are also studies of greater evidence such as meta-analyses (for example: Souza BDM, Dutra KL, Kuntze MM, Bortoluzzi EA, Flores-Mir C, Reyes-Carmona J, Felippe WT, Porporatti AL, De Luca Canto G. Incidence of Root Resorption after the Replantation of Avulsed Teeth: A Meta-analysis. J Endod. 2018 Aug;44(8):1216-1227. doi: 10.1016/j.joen.2018.03.002. Epub 2018 Jun 1. PMID: 29866405.) that should be taken into account when discussing the topic in the Introduction. 

The Material and method is well written and structured although it does not appear therein that informed consent should have been obtained from the patients or from their parents or legal guardians if they were minors. 

The presentation of the cases is correct although I miss some clinical photographs to evaluate other factors that should also be taken into account for the evaluation of the cases. 

The DISCUSSION and conclusions are correct.

Round 2

Reviewer 2 Report

The article has substantially improved.
